# Long-Term Results of Endovascular Treatment with Nitinol Stents for Femoropopliteal TASC II C and D Lesions

**DOI:** 10.3390/medicina58091225

**Published:** 2022-09-05

**Authors:** Michaela Kluckner, Patrick Nierlich, Wolfgang Hitzl, Thomas Aschacher, Alexandra Gratl, Sabine Wipper, Manuela Aspalter, Herve Moussalli, Klaus Linni, Florian K. Enzmann

**Affiliations:** 1Department of Vascular Surgery, Medical University Innsbruck, A-6020 Innsbruck, Austria; 2Chirurgie Nierlich, Vienna Private Hospital, A-1090 Vienna, Austria; 3Department of Ophthalmology and Optometry, Paracelsus Medical University Salzburg, A-5020 Salzburg, Austria; 4Research and Innovation Management, Paracelsus Medical University, A-5020 Salzburg, Austria; 5Research Program Experimental Ophthalmology and Glaucoma Research, Paracelsus Medical University, A-5020 Salzburg, Austria; 6Department of Cardiovascular Surgery, Clinic Floridsdorf and Karl Landsteiner Institute for Cardio-Vascular Research, A-1210 Vienna, Austria; 7Department of Cardiac, Vascular and Endovascular Surgery, Paracelsus Medical University, A-5020 Salzburg, Austria

**Keywords:** femoropopliteal lesion, nitinol stent, peripheral arterial disease, TASC

## Abstract

*Background and Objectives*: The feasibility of endovascular treatment (EVT) for Trans-Atlantic Inter-Society Consensus (TASC) II C and D femoropopliteal artery lesions has been described, but no prospective study has performed a long-term follow-up. The aim of this study was to report the long-term results of nitinol stents (NS) for the treatment of long femoropopliteal lesions. *Materials and Methods*: A single-center prospective, randomized controlled trial (RCT) comparing EVT with NS and vein bypass surgery was previously performed. The EVT group’s follow-up was extended and separately analyzed with primary patency as the primary endpoint. The secondary endpoints were technical success, secondary patency, reinterventions, limb salvage, survival, complications, and clinical improvement. *Results*: Between 2016 and 2020, 109 limbs in 103 patients were included. A total of 48 TASC II C and 61 TASC II D lesions with a mean lesion length of 264 mm were reported. In 53% of limbs, the indication for treatment was chronic limb-threatening ischemia. The median follow-up was 45 months. Technical success was achieved in 88% of cases, despite 23% of the lesions being longer than 30 cm (retrograde popliteal access in 22%). At four-year follow-up, primary patency, secondary patency, and freedom from target lesion revascularizations were 35%, 48%, and 58%, respectively. Limb salvage and survival were 90% and 80% at 4 years. Clinical improvement of at least one Rutherford category at the end of follow-up was achieved in 83% of limbs. *Conclusions*: This study reports the longest follow-up of endovascular treatment with nitinol stents in femoropopliteal TASC II C and D lesions. The results emphasize the feasibility of an endovascular-first strategy, even in lesions beyond 30 cm in length, and clarify its acceptable long-term durability and good clinical outcomes. Large multicenter RCTs with mid- and long-term follow-up are needed to investigate the role of different endovascular techniques in long femoropopliteal lesions.

## 1. Introduction

Atherosclerotic disease is often generalized and patients with lower extremity peripheral arterial disease (PAD) have an increased risk of cardiovascular mortality and morbidity due to arterial stenoses and occlusions [1]. The ankle–brachial index (ABI) is the main non-invasive diagnostic as well as surveillance tool for PAD, and values ≤ 0.90 or >1.40 are associated with elevated rates of mortality and coronary events. Its sensitivity is worse in patients with diabetes, a major risk factor for PAD, due to the medial calcification of the arteries [2]. Diabetic patients have an increased risk of major amputation with multi-level PAD with more severe below-the-knee involvement and reports of higher rates of restenosis after revascularization [3,4].

Femoropopliteal artery lesions account for approximately half of all PAD interventions [5]. Over the past decades, endovascular treatment (EVT) has become the first-line treatment for most PAD pathologies, which is reflected in PAD guidelines. While the Trans-Atlantic Inter-Society Consensus (TASC II) document recommended EVT for femoropopliteal lesions shorter than 15 cm [6], the European Society for Vascular Surgery (ESVS) extended this recommendation to all lesions shorter than 25 cm in their current PAD guidelines. For longer pathologies, both recommend vein bypass surgery as the primary treatment option [7], which is mainly based on the BASIL trial published in 2005 [8].

Endovascular techniques are constantly evolving and especially the introduction of nitinol stents (NS), drug-eluting balloons (DEB), and drug-eluting stents (DES) provided superior results compared to plain-balloon angioplasty (PBA) [9,10,11]. The aim of all adjuncts to PBA is to prevent restenosis as a result of the vascular injury. Mechanical stretching, subintimal hemorrhage, and endothelial denudation of the artery activate inflammatory responses leading to restenosis [4]. Temporarily, the meta-analysis of Katsanos et al. limited the application of paclitaxel-eluting devices in femoropopliteal arteries in many centers, as they reported an increased risk of death after their use [12]. Nevertheless, data suggest that newer generations of bare NS provide good patency rates in long femoropopliteal lesions [13,14], also in comparison to DES [15].

A general issue is that the majority of endovascular studies report the treatment of lesions <15 cm (TASC II A and B), presenting mostly short-term results. Recently published studies by Böhme et al. and Nasr et al. on the treatment of femoropopliteal lesions >15 cm (TASC II C and D) using heparin-bonded stent grafts and interwoven NS with two- and three-year results are both limited by small patient cohorts [14,16].

Between 2016 and 2020, our group conducted the largest randomized controlled trial (RCT) comparing PBA with NS and vein bypass surgery for the treatment of long femoropopliteal lesions [17]. The aim of the current study was to focus on the endovascular group of the RCT and extend the follow-up of this cohort to report long-term results of NS for the treatment of TASC II C and D lesions.

## 2. Materials and Methods

This study included all patients treated in the endovascular group of a single-center, prospective RCT comparing PBA and NS with primary vein bypass surgery in femoropopliteal TASC II C and D lesions [17]. For the present study, the follow-up of those patients was extended. The study was conducted in accordance with the Declaration of Helsinki and the principles of Good Clinical Practice guidelines at a high-volume university vascular center. It was approved by the Ethics Committee Salzburg (415-E/1938/3-2015) and registered at ISRCTN.com (ISRCTN18315574).

The primary endpoint was primary patency. The secondary endpoints were technical success, primary assisted patency, secondary patency, freedom from target lesion revascularization (TLR), limb salvage, survival, local complications (distal embolization, minor amputation, pseudo aneurysm, vessel perforation), systemic complications (congestive heart failure, myocardial infarction, postinterventional anemia, renal impairment, sepsis, stroke), and clinical improvement (Rutherford category—change [18]), as previously defined [19].

Only patients with severe intermittent claudication (<200 m of walking distance) as well as critical limb threatening ischemia (CLTI), being defined by rest pain or ischemic lesions (Rutherford categories 3–6) were included. The femoropopliteal TASC II type C or D lesions were identified with computed tomographic angiography, digital subtraction angiography, or magnetic resonance angiography. Patients were excluded if they were too frail for open surgery (American Society of Anesthesiologists classification—ASA > 3), had chronic kidney disease (glomerular filtration rate <45 mL/min/1.73 m^2^) or vasculitis, as previously described [19].

Technical success was defined as the completion of the intervention with a residual stenosis of less than 30%. The absence of flow-limiting stenosis (peak systolic velocity (PSV) ratio > 2.5) or occlusion of the treated artery was defined as primary patency. Secondary patency was determined as a secondary intervention performed for stent occlusion in a subsequently patent vessel. Reintervention due to a flow-limiting stenosis or occlusion in the previously treated segment as well as clinical deterioration was defined as TLR. The change of Rutherford categories was used to determine clinical improvement [18].

Preoperatively, patients received 100 mg acetylsalicylic acid, unless oral anticoagulation was indicated. For the endovascular interventions, contralateral retrograde, ipsilateral antegrade, or distal access via puncture of the distal popliteal artery, in a prone-positioned patient, was used per physician discretion followed by systemic heparinization. Either endoluminal or subintimal passage was attempted to cross the femoropopliteal lesions. Re-entry devices were used (Outback catheter, Cordis, Milpitas, CA, USA) if necessary. Standard balloon pre-dilatation was used (1 mm smaller than the reference vessel diameter), followed by NS with 1 mm oversizing (Facile, amg International, Winsen, Germany; Pulsar-18, Biotronik, Berlin, Germany). Five-millimeter overlaps were performed in cases where more than one stent was deployed. Post-dilatation with PBA using equal diameters as the stents was used. No drug-eluting, atherectomy, or scoring devices were used. Only flow-limiting dissections or residual stenoses of more than 30% were stented in popliteal artery and common femoral artery lesions. The standard antiplatelet therapy was clopidogrel 75 mg for 6 weeks in addition to preoperative medication. Patient follow-up was performed in an outpatient setting at 2 and 4 weeks, 3, 6, and 12 months, and per physician discretion thereafter including ABI, clinical examination, and ultrasound, as previously described [19].

For the statistical analysis data, consistency was checked and data were screened for normality (Kolmogorov–Smirnov test) and outliers by two independent reviewers. Independent Student’s t-tests (if data were deviating from normality, bootstrap *t*-tests) were used to compare different patient groups. Crosstabulation tables were tested using Pearson’s chi-squared and Fisher’s Exact test. For primary, primary assisted, and secondary patency, limb salvage, survival, and freedom from TLR Kaplan–Meier analyses were done and tested by using a log-rank test with equal weighting. Standard errors and corresponding numbers at risk are given in Kaplan–Meier plots to illustrate the results.

Univariate and multivariable Cox regression models were used to analyze various covariates. All primary and secondary endpoints were tested with a significance level of 5%. The statistical analyses in this report were performed with STATISTICA 13 (StatSoft, Tulsa, OK, USA) and NCSS (NCSS 10, NCSS, LLC. Kaysville, UT, USA).

## 3. Results

### 3.1. Patient Characteristics

A total of 1469 patients were screened for inclusion in the study between 2016 and 2020, but the majority (1260) had to be excluded due to ipsilateral iliac artery lesions (*n* = 392), missing consent (*n* = 201), previous ipsilateral bypass surgery (*n* = 179), embolic occlusion (*n* = 129), ASA classification > 3 (*n* = 84), chronic kidney disease without requiring dialysis (*n* = 74), acute limb ischemia (*n* = 62), hybrid procedures (*n* = 61), use of a prosthetic bypass conduit (*n* = 55), and no patent tibial arteries (*n* = 23). The remaining 209 patients with 218 femoropopliteal TASC II C and D lesions were randomized for vein bypass surgery or EVT [17]. A total of 103 patients with 48 TASC II C and 61 D lesions were included in the current study. The extended median follow-up was 45 months (IQR: 24–55) and only two patients were lost to follow-up after 49 and 54 months.

The demographic data, cardiovascular risk factors, ASA classification, preoperative ABI, and Rutherford categories are shown in Table 1. Lesion and procedural details are depicted in Table 2. The TASC II C lesions were mostly located in the SFA, with only two cases of CFA stenosis and 10 lesions affecting the popliteal artery (P1 and P2). Of the TASC II D lesions, 59 were total occlusions of the SFA, 37 lesions extended into the popliteal artery (P1: 16; P2: 12; P3: 9), and 24 presented with complete occlusions of the popliteal artery.

The mean lesion length was 264 mm ± 58 (lesion >15 cm: *n* = 19; >20 cm: *n* = 29; >25 cm: *n* = 36; >30 cm: *n* = 25). Due to mostly 6 to 8 cm long NS, the mean number of used stents was four (one stent: *n* = 5; two stents: *n* = 10; three stents: *n* = 10; four stents: *n* = 24; five stents: *n* = 20; six stents: *n* = 23; seven stents: *n* = 4).

### 3.2. Primary Endpoint

The primary patency at 1, 2, and 4 years was 59%, 50%, and 35%, respectively (Figure 1). Univariate and multivariable Cox regression models for the loss of primary patency showed that distal embolization and Rutherford Category 5 and 6 are the biggest risk factors, as depicted in Table 3.

### 3.3. Secondary Endpoints

Of the 109 TASC II C and D lesions, 96 (88%) could be crossed, despite 23% of the lesions being longer than 30 cm. Of the remaining 13 lesions (12%) that were treated unsuccessfully, most were TASC II D lesions (92%) with chronic total occlusions (CTO) (92%) and a mean length of 302 ± 125 mm. Seven of the thirteen lesions were subsequently treated with bypass surgery. Five of the bypasses remained patent during follow-up, and the other two had major amputation. Four of the thirteen legs clinically stabilized or improved with conservative treatment, so no TLRs were performed during follow-up. Endovascular re-interventions after 11 and 12 months were performed to treat the remaining two lesions. The only factors significantly associated with technical failure in a Cox regression analysis were: lesion length (*p* = 0.01), one stenosis-free outflow vessel (*p* = 0.04), and TASC II D lesion (*p* = 0.005).

The primary assisted patency after 1, 2, and 4 years was 69%, 57%, and 42%, respectively. At 1, 2, and 4 years, the secondary patency was 72%, 62%, and 48% (Figure 2). Univariate and multivariable Cox regression models showed that more than one stenosis-free outflow vessel was the only significant factor reducing the risk of loss of secondary patency, as displayed in Appendix A.

Freedom from TLR was 75%, 73%, and 58% at 1, 2, and 4 years, respectively (Figure 3). Distal embolization was the biggest risk factor for TLR in the univariate and multivariable Cox regression models, while popliteal stenting was only significant in the univariate analysis, see Table 4. A total of 86 revascularizations were performed after the index-interventions (53 redo-angioplasties, 20 bypass procedures, nine surgical thrombectomies, and four catheter-directed thrombolyses). Due to clinical improvement despite restenosis (*n* = 4) or reocclusion (*n* = 15) of the treated lesions, patients did not receive further interventions during follow-up.

The limb salvage rates were 99%, 96%, and 90% at 1, 2, and 4 years, respectively (Figure 4). A total of nine major amputations were performed during follow-up. No procedure-related deaths or 30-day mortality was documented. Survival at 1, 2, and 4 years was 90%, 88%, and 80%, respectively. The overall number of complications was 45, with a majority of local complications (*n* = 35, Appendix A). Fasciotomy was necessary after 2 of 10 vessel perforations due to crural compartment syndrome. In three of the seven cases of distal embolization, urgent catheter-directed thrombolysis had to be performed. The mean in-hospital stay was 3 ± 3.5 days.

Clinical improvement at the end of follow-up was detected in the majority of patients (83%—at least one Rutherford category). The mean change of Rutherford categories was 2 ± 1.5 (SD) after EVT and 77% of patients had either no symptoms or only moderate claudication (Rutherford 0: 21%, Rutherford 1: 29%, Rutherford 2: 27%). The ABI increased from a mean pre-procedural 0.52 ± 0.21 to 0.73 ± 0.22 at the end of follow-up. In patients with tissue loss, wound healing was achieved in 35 (85%). The mean time to wound healing was 5.3 ± 3.8 months.

## 4. Discussion

The role of EVT in the femoropopliteal artery segment has changed considerably over the past two decades [20]. Technical advances have made it possible to treat more complex lesions and extend the indication for endovascular interventions [7]. The current study reports the longest follow-up of EVT using PBA and NS for femoropopliteal TASC II C and D lesions, providing data on its durability and clinical outcomes in claudicants and CLTI patients.

As described previously, EVT for femoropopliteal TASC II C and D lesions has several advantages in comparison to bypass surgery: shorter procedure length, shorter in-hospital stays, less morbidity, and faster recovery [17,21]. We recently showed in an RCT comparing NS with vein bypass in TASC II C and D lesions that primary patency, freedom from TLR, and limb salvage were not significantly different. Vein bypass was only superior in secondary patency and change in Rutherford categories [17]. Other groups had equal findings with very similar patency rates for bypass surgery and EVT, despite different endovascular devices, smaller groups, and shorter follow-up [21,22,23].

Comparing studies investigating the outcome of EVT for long femoropopliteal lesions is challenging. Cohorts differ in lesion characteristics, patient risk factors, and follow-up time. Most studies reported only one- or two-year results with patient numbers frequently below 100 or even 50. Similarly heterogeneous were the lesions treated with mean lengths between 117 and 252 mm and CTO rates between 42% and 93%. While some studies only included claudicants, others focused on CLTI patients [13,14,24,25,26,27,28,29,30].

Considering these aspects, there still remain studies comparable to our current one. For the STELLA PTX registry, 12-month results utilizing DES in 48 limbs with a mean lesion length of 252 mm were reported. The primary and secondary patency were similar, with 53% and 80% vs. 59% and 72% in our patients [25]. The 30-month results of the STELLA registry, using PBA and NS as well, included 62 TASC II C (63%) and D (37%) lesions with a mean length of 220 mm and CLTI in 60% of cases. The primary and secondary patency at 30 months were 62% and 77% compared to 41% and 59% in our series. This may be attributed to the shorter lesions treated and the exclusion of all re-stenotic lesions [31].

Interestingly, a propensity-score matched analysis of these two registries showed no benefits for Paclitaxel eluting stents in terms of clinical and morphological outcomes for TASC II C/D lesions compared to bare NS [15].

The recently published 2-year data of the STELLA SUPERA trial reported 49 lesions (65% TASC II D) with a mean lesion length of 234 mm (78% CTOs) treated with PBA and interwoven NS. The primary patency and freedom from TLR at 2 years was 78% and 87% compared to 50% and 73% in our cohort. The main reasons for the differences may be the shorter lesion lengths and lower PAD severity with 71% claudicants. Unfortunately, none of the STELLA registries reported data on lesion calcification [14].

The 2-year results of the SFA-Long Study by Micari et al. included 105 patients with a mean lesion length of 251 mm treated with DCB and bail-out stenting in only 11% of cases. At 2 years, the primary and secondary patency were 70% and 80% vs. 50% and 62% in our study. However, 90% of their patients were claudicants, the CTO rate was just under 50%, and only 13% were severely calcified lesions [27]. In contrast, Torres-Blanco et al. exclusively reported TASC II D lesions in CLTI patients, which may have impaired their technical success rate (83%), primary patency (40%) and freedom from TLR (41%) at 2 years [26].

Three-year results of heparin-bonded stent grafts (Viabahn 25 cm Trial) were reported by Böhme et al. with very promising primary (53%) and excellent secondary patency (97%) rates. Their mean lesion length was almost identical to the current study at 265 mm; 93% of patients had CTOs, but also no lesion calcification was stated. They enrolled 71 patients with 91% claudicants and no Rutherford categories 5 or 6, suggesting a favorable lesion selection [16].

As stated above, long-term results on EVT are scarce, highlighting the importance of the available data. Dake et al. reported 5-year results of an RCT evaluating DES and PBA with provisional NS in 479 patients. Again, mainly claudicants (>90%) were included and the mean lesion lengths were just 63 mm and 66 mm, respectively. Consequently, a comparison of their primary patency at 4 years with 67% (DES) and 46% (PBA + provisional NS) with our current data is difficult. They concluded that DES provided better long-term patency and clinical results compared to the control group [11]. To our knowledge, this long-term benefit of DES has not been described for long femoropopliteal lesions.

The high percentage of claudicants in most studies investigating EVT for long femoropopliteal lesions limits the generalizability to all patients with TASC II C and D lesions. Our initial RCT was designed to include the same number of claudicants and CLTI patients, which is, in our opinion, more representative of the general PAD patient population [17]. This almost 50/50 distribution can also be seen in an all-comers study for EVT in long lesions [29]. In the Cox regression analysis of our study, tissue loss (Rutherford category 5 and 6) was a significant risk factor for the loss of primary patency, highlighting the relevance of also including those patients in trials. CLTI was also previously reported to be a major risk factor for restenosis [30].

Interestingly, lesion lengths of more than 25 cm or 30 cm were not significant risk factors for the loss of primary patency, secondary patency, or TLR. So, at least in our patient cohort, the importance of lesion length alone was not that high, which makes it very tempting to recommend a non-selective endovascular-first approach for all femoropopliteal lesions. This might be sensible for claudicants and patients with ischemic rest pain, but the clinical effectiveness of EVT for tissue loss remains debatable. Poor symptom relief, delayed wound healing, or even the progression of tissue loss and infection may be the consequence of incomplete revascularization [32]. Moreover, in our RCT, despite similar patency rates, clinical improvement after vein bypass was superior compared to the endovascular group [17]. The consequences of failed EVT on subsequent open surgical procedures is also an important point. There is evidence suggesting inferior bypass results after failed EVT compared to primary bypass procedures [33,34]. In the current study, distal embolization during EVT was the most significant risk factor for the loss of primary patency as well as TLR, highlighting the importance of lesion selection and preventing technical errors.

These findings, in accordance with the acceptable long-term results in the current trial, led to an adaptation of our clinical practice towards an endovascular-first strategy for most long femoropopliteal lesions, independent of lesion length, reserving primary bypass surgery for patients with tissue loss (Rutherford 5 or 6).

### Study Limitations

The current study is a single-center study with a very selective patient cohort due to various inclusion and exclusion criteria. As a result, the conclusions drawn from these data may not be translatable to all patients with femoropopliteal TASC II C and D lesions. The separate inclusion of bilateral lesions may be a source of bias. The use of several overlaps with a mean number of four stents per lesion may also have influenced the reported patency rates in this study. Despite the significant number of diabetic patients, the toe–brachial index was not used in this study.

## 5. Conclusions

This study reports the longest follow-up of endovascular treatment with nitinol stents in femoropopliteal TASC II C and D lesions. The results emphasize the feasibility of an endovascular-first strategy, even in lesions beyond 30 cm in length, and clarify its acceptable long-term durability and good clinical outcomes. Large multicenter RCTs with mid- and long-term follow-up are needed to investigate the role of different endovascular techniques in long femoropopliteal lesions.

## Figures and Tables

**Figure 1 medicina-58-01225-f001:**
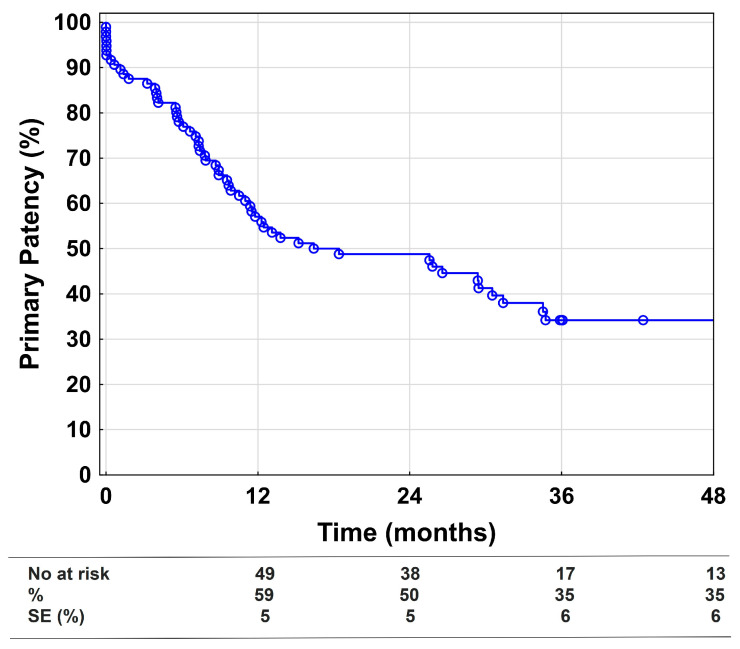
Primary patency. Kaplan–Meier estimates presenting primary patency during the 48-month follow-up, including all technically successful cases (*n* = 96). SE = standard error.

**Figure 2 medicina-58-01225-f002:**
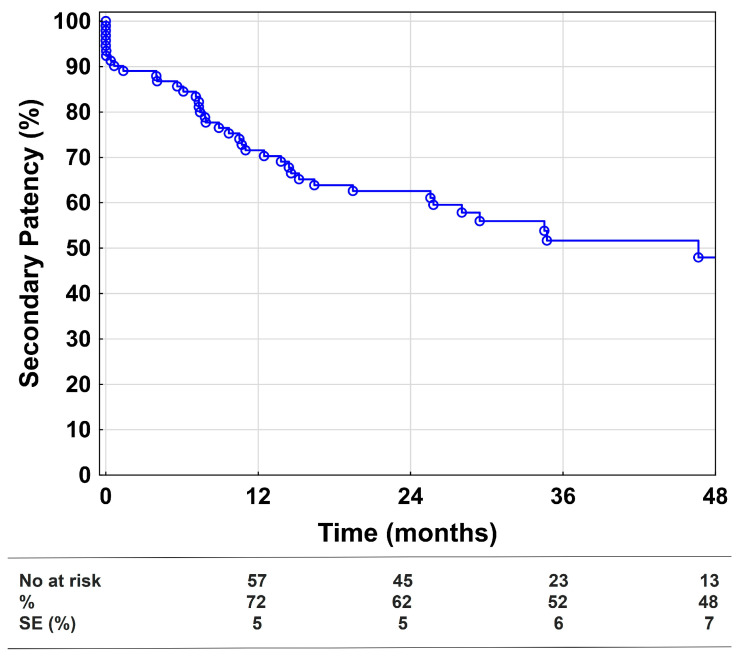
Secondary patency. Kaplan–Meier estimates presenting secondary patency during the 48-month follow-up, including all technically successful cases (*n* = 96). SE = standard error.

**Figure 3 medicina-58-01225-f003:**
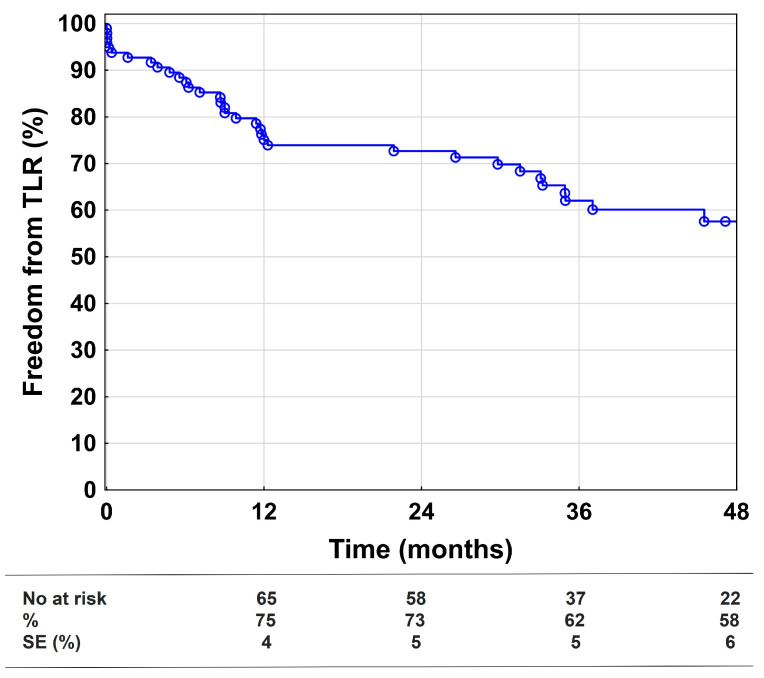
Freedom from target lesion revascularization. Kaplan–Meier estimates presenting freedom from target lesion revascularization (TLR) during the 48-month follow-up, including all technically successful cases (*n* = 96). SE = standard error.

**Figure 4 medicina-58-01225-f004:**
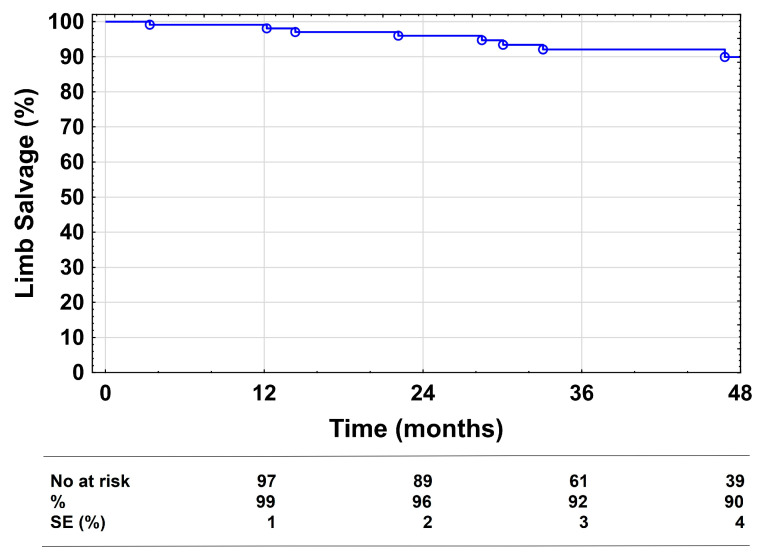
Limb salvage. Kaplan–Meier estimates presenting limb salvage during the 48-month follow-up, including all cases. SE = standard error.

**Table 1 medicina-58-01225-t001:** Patient characteristics.

	*N* = 103
Age (years)	69.3 ± 7.0
Male	69 (67%)
Body mass index, kg/m^2^	26.5 ± 4.5
Smoker	36 (35%)
Hypertension	89 (86%)
Dyslipidemia	63 (61%)
Diabetes mellitus	36 (35%)
Coronary artery disease	48 (47%)
Previous stroke	12 (12%)
Atrial fibrillation	15 (15%)
Hemodialysis	3 (3%)
ASA I	1 (1%)
ASA II	25 (24%)
ASA III	77 (75%)
ABI	0.51 ± 0.21
Rutherford category	
3	51 (47%)
4	17 (15%)
5	39 (36%)
6	2 (2%)

Values are *n* (%) or mean ± standard deviation. ABI = ankle–brachial index. ASA = American Society of Anesthesiology.

**Table 2 medicina-58-01225-t002:** Lesion and procedural details.

	*N* = 103, 109 Lesions
Lesion length (mm)	264 ± 58
TASC II C	48 (44%)
TASC II D	61 (56%)
Chronic total occlusion	87 (80%)
Recurrent lesion	31 (28%)
Reference vessel diameter (mm)	5.0 ± 0.7
Severe calcification	30 (28%)
Stenosis-free outflow vessels	
1	26 (24%)
2	53 (49%)
3	30 (27%)
Procedural length (min)	72 ± 30
Contrast agent (ml)	171 ± 80
Number of stents	4 ± 1.6
Stent diameter (mm)	6.5 ± 0.5
Stented lesion length (mm)	248 ± 98
CFA angioplasty	4 (4%)
Popliteal stenting	30 (28%)
Re-entry device used	11 (10%)
Trans-popliteal access	24 (22%)

Values are *n* (%) or mean ± standard deviation (SD). CFA = common femoral artery; TASC = Trans-Atlantic Inter-Society Consensus.

**Table 3 medicina-58-01225-t003:** Cox regression analysis of factors associated with loss of primary patency.

Loss of Primary Patency	Unadjusted HR (95% CI)	*p*-Value	Adjusted * HR (95% CI)	*p*-Value
TASC II D lesion	0.61 (0.36–1.03)	0.07		
Chronic total occlusion	1.00 (0.53–1.89)	0.99		
Lesion length > 25 cm	0.91 (0.54–1.55)	0.74		
Lesion length > 30 cm	0.88 (0.43–1.80)	0.73		
Previous intervention	1.28 (0.71–2.28)	0.41		
Reference vessel diameter < 5 mm	0.85 (0.50–1.47)	0.57		
Severe calcification	0.95 (0.51–1.77)	0.88		
Stenosis-free outflow vessels >1	1.17 (0.56–2.43)	0.67		
Rutherford cat. 5 or 6	2.26 (1.19–4.28)	0.01	2.46 (1.24–4.85)	0.01
No. of stents > 4	1.13 (0.67–1.90)	0.66		
Stent diameter > 6 mm	0.87 (0.51–1.50)	0.62		
Stent oversizing > 1 mm	1.02 (0.59–1.77)	0.95		
Popliteal stenting	1.05 (0.59–1.88)	0.87		
Re-entry device used	0.90 (0.39–2.10)	0.80		
Trans-popliteal access	0.89 (0.48–1.67)	0.72		
Distal embolization	3.83 (1.51–9.68)	0.01	6.22 (2.31–16.79)	<0.001
>1 local complication	2.41 (0.86–6.75)	0.09		

* Variables identified in univariate analysis (*p* ≤ 0.15) were included in a multivariable Cox proportional hazards model for associations with loss of primary patency.

**Table 4 medicina-58-01225-t004:** Cox regression analysis of factors associated with target lesion revascularization.

Target Lesion Revascularization	Unadjusted HR (95% CI)	*p*-Value	Adjusted * HR (95% CI)	*p*-Value
TASC II D lesion	0.88 (0.44–1.71)	0.68		
Chronic total occlusion	1.86 (0.89–3.92)	0.10		
Lesion length > 25 cm	1.12 (0.57–2.21)	0.73		
Lesion length > 30 cm	1.69 (0.76–3.75)	0.20		
Previous intervention	1.78 (0.88–3.59)	0.11		
Reference vessel diameter < 5 mm	1.12 (0.56–2.24)	0.75		
Severe calcification	1.00 (0.45–2.20)	0.99		
Stenosis-free outflow vessels > 1	1.02 (0.52–2.00)	0.96		
Rutherford cat. 5 or 6	0.75 (0.35–1.62)	0.47		
No. of stents > 4	1.87 (0.94–3.74)	0.08		
Stent diameter > 6 mm	0.67 (0.33–1.34)	0.26		
Stent oversizing > 1 mm	0.90 (0.45–1.80)	0.77		
Popliteal stenting	2.18 (1.09–4.35)	0.03		
Re-entry device used	1.08 (0.38–3.05)	0.89		
Trans-popliteal access	0.79 (0.36–1.77)	0.57		
Distal embolization	6.37 (2.43–16.70)	<0.001	6.50 (2.17–19.44)	0.001
>1 local complication	2.88 (0.88–9.50)	0.08		

* Variables identified in univariate analysis (*p* ≤ 0.15) were included in a multivariable Cox proportional hazards model for associations with target lesion revascularization.

## Data Availability

The data presented in this study are available on request from the corresponding author.

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
