# Peer review of "Long-Term Results of Endovascular Treatment with Nitinol Stents for Femoropopliteal TASC II C and D Lesions"

_medicina, 2022, doi:10.3390/medicina58091225_

Round 1

Reviewer 1 Report

I want to congratulate the authors for this prospective monocentric study reporting the mid and long term outcomes of NS in the treatment of femoropopliteal TASC C and D lesions. The topic is interesting and the text is well written and very clear. Figures are useful and are a good complement for the text. However, there are some general points that should be discussed:

-       Authors should define a single primary endpoints. The primary endpoint should be the primary patency taking into account that the aim of the study is the long term outcomes of the endovascular treatment 

-       Line 126 : authors reported that popliteal lesions were stented only in case of flow-limiting dissections or residual stenosis. Authors should give more details about the lesion location (proximal SFA, mid SFA, distal SFA, P1, P2 or P3). Authors could not conclude about the NS patency in the popliteal region if they did not treat all the popliteal lesions with NS. 

-       Table 2 : the mean lesion length was 264 mm and the stented lesion length was 248 mm. It means there is mismatch between the lesion and the segment treated which could explain the low primary patency compared to the other studies. As far as, a mean of 4  stents were used for 248 mm stented lesion. It means that authors used stents of 60 mm of length with 4 overlaps. Why they did not use a single stent of 250 mm or only 2 long stents to avoid overlapping ? It is known that overlapping increases the risk of stenosis and occlusion.

-         Authors reported results of PP, secondary patency… at 4 years. However, the number of subject at risk at 48 months is only 13 (<10% of the starting population). Outcomes should be reported at 3years follow-up ad not 4 years because of the small number of patients followed-up at 4 years. 

-       The PP is 59%, and 50% at 1 and 2years, respectively. Authors defend this low PP rate by the presence of a high rate of CLI compared to the other studies in the literature. Authors should perform a sub-group analysis. They have to report and compare the PP of the claudicants and CLI groups.

-       Authors reported a high rate of freedom from TLR for a low rate of PP. How can the authors explain this difference? Authors should report the rate of patients with restenosis and occluded stents who have not been treated. 

-       The mean in-hospital stay was 3 ± 3.5 days. This rate is high for an endovascular treatment knowing that the majority of endovascular femoropopliteal lesions could be treated as ambulatory cases. Can the authors give an explanation?  

Reviewer 2 Report

Atherosclerotic cardiovascular disease is one of the most important problems for public health worldwide. Peripheral arterial disease is, next to coronary heart disease and cerebrovascular disease, a clinical manifestation of atherosclerotic cardiovascular disease. Endovascular procedures play the important role in the treatment of cardiovascular diseases, including peripheral arterial disease. So, scientific efforts to improve the current possibilities in the endovascular treatment of peripheral arterial disease are very important. Kluckner et al. prepared an interesting article, which gives a significant contribution to improving the current knowledge in this area. Although the article is generally well prepared and, in my opinion, it should be considered for publication in Medicina, some changes are necessary to further improve the quality and attractiveness of this paper.

1) A colon should not be placed after the word “Abstract” (line 18), as well as after subtitles of parts of the abstract, such as “Background and Objectives”, “Materials and Methods”, “Results”, “Conclusions” (lines 19, 24, 30, 39). It should not be that every part of the abstract is written from a new paragraph.

2) In my opinion, keywords should include only nouns, not adjectives. So, please consider changing the word “femoropopliteal” into an expression including a noun, such as for example “femoropopliteal disease”, “femoropopliteal stenting”, or “femoropopliteal lesion”.

3) The introduction must be improved. An additional part is necessary, in which some general information about the peripheral arterial disease will be described as well as the ankle-brachial index (ABI) should be indicated as the main diagnostic tool. The definition of peripheral arterial disease should be given. It would be worth mentioning that the clinical course of peripheral arterial disease in patients with diabetes is different in some aspects from this in patients without diabetes. In patients with diabetes arteries below the knee are more often affected as well as multilevel stenosis and occlusion are more often observed. Moreover, ABI has a lower diagnostic value in patients with diabetes, and the assessment should be completed by measurement of the toe-brachial index (TBI). Some information about restenosis should be given. Restenosis is the main problem, which diminishes the efficacy of the endovascular treatment and it may lead to the necessity of reintervention. The main mechanisms responsible for the pathogenesis of restenosis should be mentioned. (doi.org/10.3390/ijerph17249339; doi.org/10.3390/ijerph182211970; doi.org/10.3390/ijms22073601; doi.org/10.1016/j.hlc.2017.10.014).

4) It should be a space between a number and the unit (15 cm, 25 cm, 1 mm, 5mm, 75 mg) (lines 53, 55, 68, 69, 122, 123, 128).

5) It should be “and drug-eluting stents (DES)” (line 60).

6) After “et al” should be a dot (line 69).

7) In line 130, there is an abbreviation “ABI”, but I did not notice this abbreviation to be explained in the earlier parts of the text. Please, explain it in the introduction according to my suggestion written above.

8) Describe, please, carefully, how the consistency with the normal distribution of the variables was tested.

9) In table 1., it should be only “N=103”, without “109 lesions”, because in this table information related to patients’ baseline characteristics is described. It may be mentioned in the main text, that there were some patients with more than one lesion. In Table 1, when the quantity is expressed as a percentage, the number must be followed by the "%" sign.

10) In my opinion, it would be better to write “1469 patients were screened for inclusion in the study between 2016 and 2020, but the majority (1260) (…)” (line 149-150), but please consider a consultation with the English philologist.

11) I believe that descriptive statistics should be modified in some items in Table 2. In my opinion, it makes no sense to present the number of stents as mean and standard deviation. It would be much more informative to indicate the number of patients in which 1 stent, 2 stents, 3 stents, etc. have been implanted. Likewise, the average length of the change also makes no sense. It is better to choose certain ranges of lesion length, e.g. less than 5 cm, from 5 to 10 cm, etc., and state how many lesions have been in each interval. The same goes for the average amount of contrast, stent diameter, and stented lesion length.

12) The title “secondary endpoints” should have the number 3.3. (line 234)

13) It would be worth mentioning to indicate in the conclusions the necessary directions for future research in this area.

14) In the references DOI numbers should be given.

Round 2

Reviewer 2 Report

The paper has been significantly improved. I recommend it for publication in its current form.